# Evaluation of Antifungal Selective Toxicity Using *Candida glabrata ERG25* and Human *SC4MOL* Knock-In Strains

**DOI:** 10.3390/jof9101035

**Published:** 2023-10-20

**Authors:** Keiko Nakano, Michiyo Okamoto, Azusa Takahashi-Nakaguchi, Kaname Sasamoto, Masashi Yamaguchi, Hiroji Chibana

**Affiliations:** 1Medical Mycology Research Center, Chiba University, Chiba 260-8673, Japan; 2School of Medicine, Niigata University, Niigata 951-8510, Japan; 3Faculty of Medicine, University of the Ryukyus, Okinawa 903-0125, Japan

**Keywords:** antimicrobial resistance (AMR), molecular target drug, specific inhibitor, drug resistant, C4-methyl sterol oxidase, methylsterol monooxygenase 1 (MSMO1), *C. albicans*, *C. parapsilosis*, non-*albicans Candida* (NAC), stealth infection

## Abstract

With only four classes of antifungal drugs available for the treatment of invasive systemic fungal infections, the number of resistant fungi is increasing, highlighting the urgent need for novel antifungal drugs. Ergosterol, an essential component of cell membranes, and its synthetic pathway have been targeted for antifungal drug development. Sterol-C4-methyl monooxygenase (Erg25p), which is a greater essential target than that of existing drugs, represents a promising drug target. However, the development of antifungal drugs must consider potential side effects, emphasizing the importance of evaluating their selective toxicity against fungi. In this study, we knocked in *ERG25* of *Candida glabrata* and its human ortholog, *SC4MOL*, in *ERG25*-deleted *Saccharomyces cerevisiae*. Utilizing these strains, we evaluated 1181-0519, an Erg25p inhibitor, that exhibited selective toxicity against the *C. glabrata ERG25* knock-in strain. Furthermore, 1181-0519 demonstrated broad-spectrum antifungal activity against pathogenic *Candida* species, including *Candida auris*. The approach of utilizing a gene that is functionally conserved between yeast and humans and subsequently screening for molecular target drugs enables the identification of selective inhibitors for both species.

## 1. Introduction

The threat of antimicrobial resistance (AMR) has progressed in parallel with the COVID-19 pandemic caused by the SARS-CoV-2 virus [1]. These resistant pathogens include drug-resistant *Candida* [1,2,3]. In recent years, drug-resistant *Candida* species other than *Candida albicans*, called non-albicans *Candida*, have become more common, especially *Candida glabrata*. Therefore, the development of new antifungal agents against them is urgently needed [4,5,6].

Ergosterol and its synthetic pathway have served as sources for antifungal drug targets [5,7]. Polyenes act directly on ergosterol [8,9], while azoles inhibit lanosterol 14α-demethylase (Erg11p) [10,11,12,13]. Allylamines target squalene monooxygenase (Erg1p); morpholines affect C14 sterol reductase (Erg24p). Additionally, there may still be potential targets [14,15]. Interestingly, it has been demonstrated that the antifungal effects of targeting the ergosterol biosynthetic pathway are not solely attributable to ergosterol depletion. Instead, they result from the accumulation of abnormal sterols due to the interruption or bypass of the pathway. Therefore, even within the same pathway, the cellular effects vary depending on which enzyme is targeted [16].

As we can observe through the example of azoles, it is crucial to take into account the emergence of drug resistance when targeting the ergosterol synthesis pathway. In the case of *C. glabrata*, the primary mechanism driving azole resistance involves the upregulation of efflux pumps, particularly *CDR1*, which is mediated by a transcription factor, *PDR1* [17,18,19,20,21,22,23,24]. Other studies have also highlighted that azole resistance coincides with stress induced by mitochondrial loss or dysfunction, leading to the activation of *PDR1*, and subsequently, the upregulation of *CDR1* [25,26,27,28,29,30]. Furthermore, as a strategy for azole resistance in *C. glabrata*, our attention has been directed towards the role of host cholesterol uptake. This process enables the uptake of cholesterol from the host and its utilization as a substitute for ergosterol [16,31,32,33,34]. In a prior investigation, we demonstrated that out of the 12 genes involved in the ergosterol pathway, only the gene knockdown of *ERG25* or *ERG26* was not compensated by serum. Additionally, we conducted a comparative analysis of the *ERG25* and *ERG26* genes with their orthologs in other fungi and humans. The results indicated that the amino acid sequence of *ERG25* is more conserved across fungi than *ERG26* and exhibits less similarity to its human ortholog [16]. These insights have led us to place a stronger focus on *ERG25*.

A cell-free assay system is useful for evaluating target-molecule-specific activity [35]. However, Erg25p has been suggested to form a complex with the ER membrane proteins Erg28p, Erg27p, and Erg26p. This complex formation has not been fully clarified [36,37], and difficulties exist in accurately reconstructing the membrane assembly in vitro. Thus, we focused on the use of a knock-in strain of *Saccharomyces cerevisiae*. In total, 47% of genes in *S. cerevisiae*, including *ERG25*, have been reported to be complemented by human genes [38]. These genes can be used for the phenotypic examination of selective toxicity using knock-in strains.

Several drugs targeting Erg25p have been reported. For example, 6-amino-2-n-pentylthiobenzothiazole (APB) has been shown to inhibit mycelial formation in *C. albicans* [39], while PF1163A and PF1163B exhibit inhibitory effects against *C. albicans* and display a slight inhibitory activity against *C. glabrata*, *C. krusei*, *C. parapsilosis*, and *Aspergillus fumigatus* [40,41]. Additionally, diazabolins have been identified as inhibitors against *S. cerevisiae* [42]. It is worth noting that *S. cerevisiae* is vulnerable to 1181-0519 (N-[(2E)-2-[(4-nitrophenyl) hydrazinylidene]propyl] acetamide) [43,44]. As far as our knowledge extends, there have been no reports to date evaluating the activity of 1181-0519 against *Candida* species. 

In this study, we demonstrated functional complementation between the genes *C. glabrata ERG25* and human *SC4MOL* by replacing *ERG25* in *S. cerevisiae*. Subsequently, we used these strains to assess the selective toxicity of an Erg25p inhibitor, 1181-0519. The use of this phenotypic in vitro evaluation system for target molecules presents significant advantages, as it has the potential to expedite and streamline drug development. Furthermore, this experimental system shows promise for evaluating the selective toxicity of Erg25p and can serve as a valuable tool for screening other inhibitors targeting either Erg25p or SC4MOL.

## 2. Materials and Methods

### 2.1. Strains, Plasmids, and Media

*Escherichia coli* ME9806 (iVEC3) (National Bio-Resource Project (NBRP), Mishima, Japan) was used as the cloning host. *Candida auris* CBS 10913, *C. tropicalis* CBS 94, *C. parapsilosis* CBS 604, and *C. krusei* CBS 573 were obtained through the ME9806 National Bio-Resource Project (NBRP), Chiba, Japan. The bacterial strains were grown in Luria Broth containing 50 µg/mL ampicillin (FUJIFILM Wako Pure Chemical Corporation, Osaka, Japan). The strains used in the present study are listed in Table 1. The YEp352-GAPII (containing *TDH3* promoter and *URA3*) and the YEp351-GAPII plasmid (containing *TDH3* promoter and *LEU2*) [45] were used to express recombinant proteins. All yeast strains were grown in YPD medium composed of 1% (*w*/*v*) Bacto Yeast Extract (Gibco, Miami, FL, USA), 2% (*w*/*v*) HIPOLYPEPTON (Nihon Pharmaceutical Co., Ltd., Osaka, Japan), and 2% (*w*/*v*) glucose or synthetic defined minimal medium (SD) (0.17% (*w*/*v*) Yeast Nitrogen Base without amino acids and ammonium sulfate, 5% (*w*/*v*) ammonium sulfate (Wako), 2% (*w*/*v*) glucose, and appropriate amino acids). The solid media were supplemented with 2% (*w*/*v*) agar (Wako). PF1163B (kindly provided by Meiji Seika Pharma Co., Ltd., Odawara, Kanagawa, Japan) and 1181-0519(N-[(2E)-2-[(4-nitrophenyl) hydrazinylidene] propyl] acetamide) (ChemDiv, San Diego, CA, USA) were used as growth inhibitors.

### 2.2. Construction of Sc(hERG25) and Sc(CgERG25)

The gene encoding *SC4MOL* (*hERG25*), which was optimized for expression in *S. cerevisiae*, was obtained from Eurofins Genomics KK (Tokyo, Japan) (Appendix A). A DNA fragment containing *hERG25* was synthesized via polymerase chain reaction (PCR) using the primers hERG25-F1 and hERG25-R1 (all primers are listed in Appendix A), and linear YEp352-GAPII containing *TDH3* promoter and the *URA3* gene was amplified using primers YEp352-GAPII-F and YEp352-GAPII-R. The DNA fragment was inserted into YEp352-GAPII by in vivo cloning using *E. coli* strain ME9806 (iVEC3) [46]. The resulting plasmid, YEp352-GAPII-hERG25, was transformed into Sc(*erg25Δ/ERG25*) cells as described previously [47]. Ura^+^ transformants were then inoculated on a solid sporulation medium containing 1% (*w*/*v*) potassium acetate and 2% (*w*/*v*) agar at 28 °C for 2 days. Spore formation was confirmed under a microscope, and spores were planted on the YPD agar medium. The resulting haploid strains were sub-cultured using the replica plating method on three types of media: YPD, SD-URA, and SD + G418 (200 µg/mL) (Appendix A). The strains that grew on all three media were designated Sc(hERG25).

A DNA fragment encoding *CgERG25* was amplified using primers CgERG25-F1 and CgERG25-R1, with the genomic DNA of *C. glabrata* as a template (Appendix A). Linear YEp351-GAPII containing *TDH3* promoter and *LEU2* was amplified using the primers YEp352-GAPII-F3 and YEp352-GAPII-R3. The DNA fragment, including the region encoding *CgERG25*, was inserted into YEp351-GAPII by in vivo cloning using *E. coli* strain ME9806 (iVEC3). The resulting plasmid, YEp351-GAPII-CgERG25, was transformed into Sc(hERG25) cells and selected on SD medium without uracil and leucine. Transformants were selected on an SD plate containing 5-fluoroorotic acid (5-FOA) without leucine to obtain Sc(CgERG25).

### 2.3. Liquid Growth Assays and Determination of IC_50_

The strains were inoculated in YPD medium and grown to saturation overnight. Growth assays were performed in 96-well flat-bottom plates in 100 µL of SD medium with or without 20 mg/L uracil or 20 mg/L leucine. Growth rates were determined by measuring the optical density at 600 nm (OD_600_) after shaking at 1000 rpm for 10 s every 10 min for at least 288 cycles at 28 °C using an Infinite M200PRO (TECAN, Männedorf, Switzerland). Drugs were dissolved in dimethyl sulfoxide (DMSO) and dispensed into the plates. The final concentration of DMSO was adjusted to 0.1% (*v*/*v*). The growth rate of each strain was calculated as follows: (1) the first 10 OD readings were averaged and subtracted from all OD readings of the corresponding curve to set the baseline of the growth curve to zero, and (2) the area under the curve (AUC) was then calculated as the sum of all OD readings. ‘Relative growth’ was calculated as previously described [48] and as follows: (AUC_condition_ − AUC_control_)/AUC_control_, where AUC_control_ represents the growth rate of the reference condition that was assayed on the same microtiter plate. Half maximal (50%) inhibitory concentration (IC_50_) is the concentration of the drug that is 50% of the OD_600_ value of a well containing only medium and 0.1% (*v*/*v*) DMSO after 48 h of culture.

### 2.4. Spotting Assay

The strains were then inoculated on solid synthetic defined (SD) medium with or without 20 mg/L uracil or 20 mg/L leucine containing 2% (*w*/*v*) agar at 28 °C for 2 days. Two to three single colonies were suspended in saline solution (Otsuka Pharmaceutical Co., Ltd., Ltd., Tokyo, Japan), adjusted to 2 × 10^6^ cells/mL using a cell counter (WATSON Co., Ltd., Tokyo, Japan), serially diluted 1:10, and spotted (5 µL) on solid SD medium containing 2% (*w*/*v*) agar using an epMotion^Ⓡ^ 96 (Eppendorf, Hamburg, Germany). After 48 h of incubation at 28 °C, pictures of the growth of cells were taken. Assays were repeated three times.

### 2.5. Determination of Minimum Inhibitory Concentration (MIC)

The determination of the minimum inhibitory concentration (MIC) of 1181-0519 was primarily conducted following the CLSI-M27A guidelines. Various strains were streaked onto YPD agar medium and incubated at 28 °C for 1 or 2 days until colonies became visible. Subsequently, five colonies with a diameter of approximately 1 mm were selected and suspended in tubes containing 1 mL of sterile saline solution (0.9% (*w*/*v*)). The resulting suspension was vigorously mixed by vortexing for 1 min. Cell counts were obtained using a blood cell counting board, and the final cell count was adjusted to 3 × 10^3^ cells/mL using Roswell Park Memorial Institute (RPMI) MOPS medium (pH 7) in all wells. The wells in the first column contained 32 µM of 1181-0519, 1 × 10^3^ cells/mL, and 0.5% (*v*/*v*) DMSO in 200 µL of RPMI MOPS. The other wells contained 1 × 10^3^ cells/mL and 0.5% (*v*/*v*) DMSO in 100 µL of RPMI MOPS. These wells were thoroughly mixed by pipetting, and then 100 µL was transferred from the wells of the first column to the adjacent wells, repeating this process to achieve a 1/2 dilution of the drug in subsequent wells. The wells in the last column did not include 1181-0519. The cells were incubated for 24 h at 28 °C. The MIC, which is the lowest concentration of the drug at which there was no visible turbidity, was determined by visually comparing the growth with the wells that had no drug.

### 2.6. Evaluation of Cytotoxicity

A431 (human cell line derived from epidermoid carcinoma) and HepG2 (human hepatoma) cell lines were obtained from RIKEN BRC and JCRB CELL BANK, respectively. All cells were cultured in RPMI-1640 medium provided by FUJIFILM Wako Pure Chemical Corporation. This medium was supplemented with 10% (*v*/*v*) fetal bovine serum from Grand Island Biological Company (Gibco), 0.1% (*v*/*v*) penicillin, and 0.1% (*v*/*v*) streptomycin, also from Gibco. The cell cultures were maintained in a humidified 5% (*w*/*v*) CO_2_ incubator at a constant temperature of 37 °C. After 24 h of incubation, both with and without the presence of 100 µM of 1181-0519, we added Premix WST-1 (Takara Bio Inc., Shiga, Japan) into each well. Subsequently, the plates were incubated for 1.5 h at 37 °C, and we measured the absorbance of the wells at both 450 nm and 600 nm using an iMark microplate reader from BIO-RAD (Hercules, CA, USA). Cell viability was determined by subtracting the absorbance value at 600 nm from the absorbance value at 400 nm, which reflects the catalytic activity of tetrazolium salt in forming formazan dye through mitochondrial dehydrogenase activity.

### 2.7. Statistical Analyses

All experiments were conducted a minimum of three times. Statistical analyses, including t-tests and one-way ANOVA tests (both two-tailed and unpaired), were performed using GraphPad Prism 10.0.3 (GraphPad Software, San Diego, CA, USA) to calculate *p*-values.

## 3. Results

### 3.1. Complementarity of hERG25 and CgERG25 in S. cerevisiae

We knocked in *ERG25* of *C. glabrata* or its human ortholog, SC4MOL, into *S. cerevisiae* and isolated clones from tetrads that lacked the *S. cerevisiae* endogenous *ERG25* (Appendix A). Consequently, we replaced the *S. cerevisiae ERG25* with either *CgERG25* or *hERG25*(*SC4MOL*), resulting in the creation of Sc(CgERG25) and Sc(hERG25) strains, respectively (Appendix A), and their transcription levels showed no relatively significant differences (Appendix A). Given the role of *ERG25* as an essential gene in *S. cerevisiae*, the growth of both Sc(hERG25) and Sc(CgERG25) demonstrated functional complementation. To evaluate the completeness of complementation by *hERG25* and *CgERG25*, we conducted an analysis of the growth curves of Sc(hERG25), Sc(CgERG25), and the control strain BY4741. As a result, their growth curves exhibited nearly identical slopes, and statistical analyses employing t-tests and one-way ANOVA indicated no significant differences (Figure 1A). Additionally, a spot assay showed no different colony formation characteristics among them (Figure 1B). Furthermore, we subjected the knock-in strains to incubation in the presence or absence of fluconazole or amphotericin B, observing no significant differences in susceptibility among the strains (Appendix A). These results collectively suggest that both *hERG25* and *CgERG25* effectively complement the function of *ERG25* in *S. cerevisiae.*

### 3.2. Drug Susceptibility of Knock-In Strains with Liquid Growth Assays

PF1163B, a known inhibitor of *C. albicans* Erg25p, distinguishes itself from PF1163A by lacking an additional hydroxyl group on its side chain (Appendix A). PF1163B exhibits a broader spectrum of activity compared to PF1163A and has been reported to have slight inhibitory effects on *C. glabrata* [40]. To confirm the usefulness of PF1163B in our study, we conducted a growth inhibition comparison involving BY4741, Sc(hERG25), and Sc(CgERG25) in the presence of PF1163B (Figure 2A). The results indicated slight inhibitory activity against all three strains, although the IC_50_ was not reached even at the highest concentration of 138 μM (Table 2). Another compound, 1181-0519 (N-[(2E)-2-[(4-nitrophenyl) hydrazinylidene] propyl] acetamide) (Appendix A), has been reported to possess inhibitory activity against Erg25p in S. cerevisiae [43,44]. Thus, we compared its growth inhibitory effects on BY4741, Sc(hERG25), and Sc(CgERG25) (Figure 2B). The IC_50_ values for BY4741 and Sc(CgERG25) were 13 µM and 3 µM, respectively, whereas for Sc(hERG25), the IC_50_ was greater than 32 µM (Table 2). Consequently, we observed growth inhibition of 1181-0519 against Sc(CgERG25) but not against Sc(hERG25).

### 3.3. Evaluation of 1181-0519 Spectrum

To assess the potential of 1181-0519 as an antifungal agent, we conducted an evaluation of its activity against *Candida* species. We employed the Clinical Laboratory Standards Institute (CLSI) M27A3 method, which is a standard approach for assessing antifungal drugs. The *Candida* strains chosen as subjects included *C. albicans* SC5314, *C. glabrata* CBS 138, *C. auris* CBS 10913, *C. tropicalis* CBS 94, *C. parapsilosis* CBS 604, and *C. krusei* CBS 573. The results indicated that 1181-0519 exhibited strong inhibitory activity against *C. albicans*, *C. glabrata*, *C. auris*, *C. parapsilosis*, and *C. krusei*, with a minimum inhibitory concentration (MIC) value of less than 2 μM, while relatively weak inhibitory activity was observed against *C. tropicalis*, with an MIC of 16 µM (Table 3).

### 3.4. Cytotoxicity of 1181-0519

To evaluate the cytotoxicity of 1181-0519, we incubated A431 and HepG2 cell lines in medium with or without 100 µM 1181-0519, and then measured mitochondrial dehydrogenase activity using the WST-1 assay with a formazan dye concentration. The absorbance of the formazan dye, serving as an indicator of cell viability, is shown in Figure 3. The results of the t-test indicated that, for both the A431 and HepG2 cell lines, the p-values were 0.227 and 0.338, respectively, suggesting no significant differences between cultures with and without 1181-0519 in the medium. Therefore, no cytotoxicity was detected with 1181-0519 even at a concentration of 100 µM.

### 3.5. Homology Analysis of Erg25p

Erg25p is a non-heme iron-requiring enzyme and is characterized by three histidine motifs that are conserved throughout eukaryotes [49]. These three histidine motifs, namely HX3 H, HX2 HH, and HX2 HH, are iron-binding sites (Appendix A) and are presumed to be important for enzymatic function [49]. The amino acid homology between human and *C. glabrata*, human and *S. cerevisiae*, and *C. glabrata* and *S. cerevisiae* was 37.5%, 34.5%, and 89.0%, respectively (EMBOSS Water < Pairwise Sequence Alignment < EMBL − EBI), and the three histidine motifs were conserved in all species (Appendix A). Therefore, they do not account for differences in susceptibility between Sc(hERG25) and Sc(CgERG25) to 1181-0519. Consequently, we can infer that the binding site of Erg25p in 1181-0519 is located outside of those motifs.

## 4. Discussion

C4-methyl sterol monooxygenase (Erg25p), which is essential for the growth of *S. cerevisiae* [50], *C. albicans* [51], and *C. glabrata* [16], represents a promising target for antifungal drugs with higher efficacy compared to Erg11p, the target molecule of azoles [16]. However, the presence of an orthologous gene for *ERG25* in humans necessitates the verification of inhibitor selectivity against fungi. We knocked in *C. glabrata ERG25* or the human orthologous gene SC4MOL into *S. cerevisiae* lacking *ERG25*, confirming their functional complementation (Figure 1). Using these knock-in strains, we demonstrated that 1181-0519, an inhibitor of Erg25p in *S. cerevisiae*, exhibited no inhibitory activity against Sc(hERG25), while it did against Sc(CgERG25) (Figure 2). Furthermore, 1181-0519 displayed broad-spectrum activity against *Candida* species (Table 3) while exhibiting no toxicity to cultured human cells A431 and HepG2 (Figure 3). These findings suggest that 1181-0519 holds potential as an antifungal candidate. Although the growth inhibitory activity of 1181-0519 against *S. cerevisiae* has been previously reported [43,44], to the best of our knowledge, this is the first report evaluating its efficacy against pathogenic fungi and human genes.

Selective toxicity against fungal cells is a crucial consideration in the development of antifungal drugs, and various methods are available for its assessment. One such method is a molecular-level assay system employing a cell-free system. However, due to the formation of a complex involving Erg25p on the endoplasmic reticulum membrane [36,37], establishing an in vitro cell-free assay system has proven to be challenging. In this study, we demonstrated that simple culture experiments can effectively evaluate the selective inhibitory activity of enzymes with identical functions in different species. As a result, this study highlights the utility of a knock-in strain system as a valuable alternative experimental approach, particularly for proteins that form complexes, where constructing a cell-free assay system poses difficulties.

Once an inhibitor has been identified, attempts should be made to modify its molecular structure to increase its potency, specificity, and broader activity in pathogens. This approach is essential not only for advancing the development of improved drugs but also for pre-empting the emergence of derivative drugs that may follow. Identifying the binding domain of the target streamlines compound development. In *ERG25*, the amino acid sequences within the three histidine motifs of the active domain were conserved between human and fungal *ERG25* (Appendix A), suggesting that the binding domain of 1181-0519 is not those motifs. Notably, *C. tropicalis* exhibited lower sensitivity compared to other *Candida* species (Table 3). These variations in susceptibility could provide insights for estimating binding sites. However, since these experiments were conducted on different species using their whole cells, they may have been influenced by intracellular environments other than Erg25p. To accurately assess Erg25p-specific sensitivity to 1181-0519, it would be advantageous to standardize conditions other than Erg25p by constructing *ERG25* knock-in strains of these species in *S. cerevisiae*. This approach would enable the precise evaluation of Erg25p-specific sensitivity to 1181-0519. Subsequently, this information could inform various applications, such as docking simulations aimed at identifying the binding site of 1181-0519 to Erg25p and enhancing its spectrum of action. Therefore, further studies should be conducted on them.

Considering potential side effects is important for the development of antifungal drugs. Although *SC4MOL*, the human ortholog of *ERG25*, is not an essential gene for growth in human cells, diseases caused by mutations in this gene have been reported [52]. These diseases result in impaired cholesterol biosynthesis, leading to microcephaly, bilateral congenital cataracts, growth retardation, psoriasiform dermatitis, immune dysfunction, and intellectual disability [53,54,55,56]. Additionally, 4,4-dimethylzymosterol, a substrate of SC4MOL that accumulates upon its inhibition, has been reported to exhibit testicular meiosis activity in mammals and plays a crucial role in regulating cumulus oophorus expansion and oocyte maturation [57,58,59]. Since these diseases and abnormalities manifest during ontogenetic processes, which are challenging to reproduce in cultured cells, using cultured cells for toxicity evaluation is not appropriate. Therefore, in the process of developing antifungal drugs, side effects are typically evaluated through animal experiments, which necessitate an evaluation system that includes pathological models. Furthermore, removing a candidate compound from the development process at this stage can result in significant losses, as previous investments would be wasted. Therefore, it is important to assess selective inhibitory activity at the molecular level in advance using a knock-in strain.

The two knock-in strains produced here can also be utilized in the search for new Erg25p inhibitors other than 1181-0519. Compounds that inhibit the growth of the Sc(CgERG25) strain, but not Sc(hERG25), are likely to act specifically against CgErg25p. Thus, the difference in susceptibility between the two strains can be employed in differentially high-throughput screening for specific inhibitors of Erg25p. In contrast, a compound that inhibits growth against the Sc(hERG25) strain, but not against Sc(CgERG25), is likely to act specifically against SC4MOL. This inhibitor could contribute to the study of SC4MOL deficiency.

Some clinical isolates of *C. glabrata* have been reported to lose the ability to synthesize ergosterol and do not grow on standard laboratory medium. However, they can grow on media containing serum or bile [16,31,32,33,34]. These strains compensate for the lack of ergosterol by uptake of host cholesterol and subsequently become resistant to azoles and polyene [33,34]. The primary issue lies in the inability to isolate these sterol-requiring strains using the standard media employed in clinical diagnostic tests, as these media lack sterols [26,30], consequently leading to stealth infections. Similar risk may arise when targeting proteins encoded by *ERG1, ERG11*, or *ERG7* [16] which allow cholesterol uptake, enabling cells to continue growing in a host. Conversely, the disfunction of *ERG25* or *ERG26* prevent the uptake of host cholesterol [16], making inhibitors of Erg25p or Erg26p effective in repressing growth, even in an environment with cholesterol. In addition, the amino acid sequence of Erg25p is more fungal-specific than that of Erg26p. Therefore, Erg25p represents an excellent target molecule in the ergosterol synthetic pathway that does not promote stealth infection by *C. glabrata*.

## 5. Conclusions

While Erg25p represents a promising antifungal drug target, its inhibitors may induce side effects in humans if they interfere with human SC4MOL. However, since SC4MOL is non-essential for human cells, and these side effects manifest during the ontogeny process, toxicity cannot be detected in experiments conducted with cultured cells. For inhibitors whose toxicity cannot be assessed in cultured cells due to similar reasons, the experimental system we have demonstrated will serve as a crucial complement to the selective toxicity evaluation system, enabling the assessment of new test compounds targeting both fungi and humans. This system provides valuable insights into the direct impact of selective toxicity for these inhibitors. In fact, the implementation of this evaluation system has revealed that 1181-0519 exhibits specific efficacy against *Candida* species.

## Figures and Tables

**Figure 1 jof-09-01035-f001:**
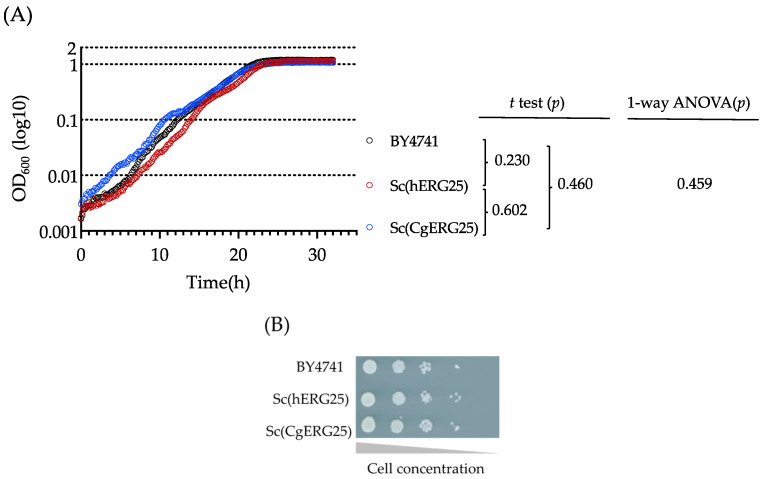
Complementation with human *SC4MOL* and *C. glabrata ERG25* in *S. cerevisiae*. (**A**) The growth curve in liquid culture of *S. cerevisiae* knock-in strains. Cells (5 × 10^4^/well) of three strains were grown in SD medium containing uracil and leucine at 28 °C, with turbidity measurements taken at OD_600_ every 10 min. The data represent the average of three replicates, and the vertical axis is presented in log_10_ notation. Statistical analyses, including t-tests and 1-way ANOVA (both two-tailed and unpaired), were performed. (**B**) Complementation assay by spotting. Starting with approximately 10^4^ cells, cells were diluted to 1/10 and spotted onto SD agar medium. The plates were subsequently placed in an incubator at 28 °C for two days.

**Figure 2 jof-09-01035-f002:**
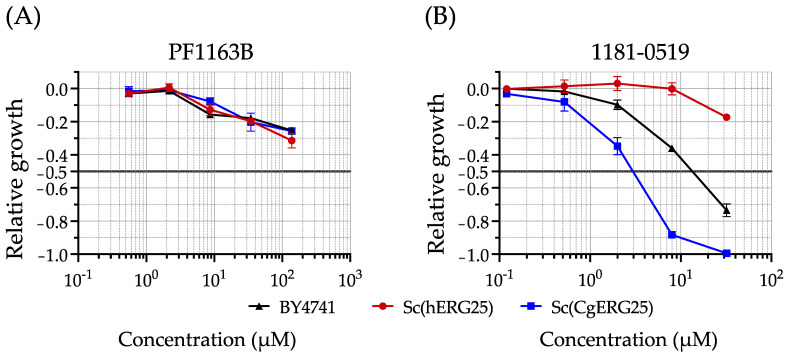
Evaluation of growth inhibition by compounds targeting Erg25p. Relative growth inhibition against knock-in strains (**A**) with PF1163B and (**B**) with 1181-0519. Sc(hERG25) and Sc(CgERG25) strains were grown in the two drugs. In both plots, the *x*-axis denotes the concentration of the respective drugs, while the *y*-axis represents the “Relative growth”, calculated as the area under the curve (AUC) relative to the absence of the drug. The data are based on the average of three replicates, with error bars indicating the standard deviation.

**Figure 3 jof-09-01035-f003:**
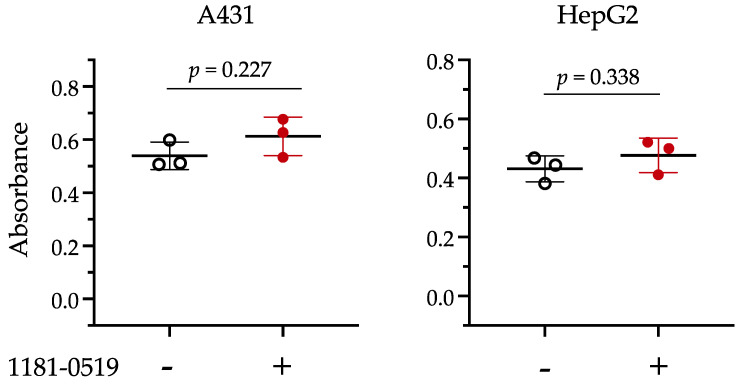
WST-1 assay for evaluation of cytotoxicity of 1181-0519. The A431 (human cell line derived from epidermoid carcinoma) and HepG2 (human hepatoma) cell lines were cultured in RPMI-1640 medium supplemented with 10% (*v*/*v*) fetal bovine serum, 0.1% (*v*/*v*) penicillin–streptomycin, and 0.1% (*v*/*v*) DMSO at 37 °C for 24 h in a 5% CO_2_ incubator. Absorbance (A_450_ − A_600_) value represents the difference between the absorbance at 450 nm minus the absorbance at 600 nm. Cells treated with 100 μM of 1181-0519 are indicated by “−” or “+”, respectively. The bars in the graph represent the average and standard deviation.

**Table 1 jof-09-01035-t001:** Strains used in this study.

Strain	Parental Strain	Genotype and (Plasmid)	Resource
Sc(erg25Δ/ERG25)	BY4743	*MATa/α his3* *Δ1/his3* *Δ1 leu2* *Δ0/leu2* *Δ0 LYS2/lys2* *Δ0 met15* *Δ0/MET15 ura3* *Δ0/ura3* *Δ0 erg25* *Δ::KanMX/ERG25*	In this study
BY4741	*S. cerevisiae*	*MATa his3* *Δ1 leu2* *Δ0 met15* *Δ0 ura3* *Δ0*	Euroscarf
Sc(hERG25)	Sc(erg25Δ/ERG25)	*erg25* *Δ::KanMX(YEp352-GAPII-hERG25)*	In this study
Sc(CgERG25)	Sc(hERG25)	*erg25* *Δ::KanMX(YEp351-GAPII-CgERG25)*	In this study
*C. albicans* SC5314		*WT*	NBRP, Chiba, Japan
*C. glabrata* CBS 138		*WT*	NBRP, Chiba, Japan
*C. auris* CBS 10913		*WT*	NBRP, Chiba, Japan
*C. tropcalis* CBS 94		*WT*	NBRP, Chiba, Japan
*C. parapsilosis* CBS 604		*WT*	NBRP, Chiba, Japan
*C. krusei* CBS 573		*WT*	NBRP, Chiba, Japan

**Table 2 jof-09-01035-t002:** Susceptibility of the strains against Erg25p inhibitors.

Strain	IC_50_ (µM)	
	PF1163B	1181-0519
BY4741	>138	13
Sc(hERG25)	>138	>32
Sc(CgERG25)	>138	3

The IC_50_ value is the concentration at which the relative growth is −0.5 in Figure 2

**Table 3 jof-09-01035-t003:** MIC of 1181-0519 against *Candida* species.

Strain	MIC (µM)
*C. albicans* SC5314	2
*C. glabrata* CBS 138	2
*C. auris* CBS 10913	1
*C. tropicalis* CBS 94	16
*C. parapsilosis* CBS 604	1
*C. krusei* CBS 573	2

MIC determination was carried out and described according to the CLSI-recommended method (CLSI M27-A3). Cell cultures at the exponential phase were diluted in RPMI-1640 medium (Sigma) to approximately 1 × 10^3^ CFU/mL and then treated with 1181-0519 at 28 °C for two days in 96-well plates; 1 µM = 0.25 µg/mL.

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
