# Peer review of "Evaluation of Antifungal Selective Toxicity Using Candida glabrata ERG25 and Human SC4MOL Knock-In Strains"

_jof, 2023, doi:10.3390/jof9101035_

Round 1

Reviewer 1 Report (Previous Reviewer 2)

It should be crrect the minor errors in the manuscript. For example, all the name species should be italic. GraphPad Prism 9.0… (Copyright information should be provided).

Author Response

Thank you for your very kind comment.
Following your support, I've italicized the species name and fixed everything else.

Best wishes,

Chibana

Reviewer 2 Report (Previous Reviewer 1)

The authors addressed all the questions pointed before. I suggest that it is accepted for publication after the following revisions:

Line 17: “deletanted” -> “deleted”

Line 18: “an Erg25p inhibitor, exhibited”: a word is missing maybe “an Erg25p inhibitor, that exhibited?”

Line 35: Polyene acts -> Polyenes act

Line 95: “YPD medium (1 % (w/v) Bacto Yeast Extract” -> something is missing, maybe: “YPD medium composed of 1 % (w/v) Bacto Yeast Extract”

Line 190- 290: From line 190 to 290, none of the species name are written in italic

Line 281: please, remove “at least”

Line 291: Please introduce the name of the human cells tested in the study

Line 305: please, remove the word “target”. In my opinion, for a reader this word is confusing. Target is more allusive to the target enzyme than to an inhibitor under study.

Line 331-333: It is difficult to understand why cultured cells cannot be used to evaluate the toxicity because of abnormalities that occur during the ontogenic processes.

Line 347: All clinical isolate of C. glabrata or some?

Line 353-356: please, rephrase. 

Line 366-368: The authors should explain why the inhibitory effect on the human enzyme cannot be detect by the viability of cultured cells. Moreover, these in vitro cell culture assays will be required any way. In other words, the authors should refer that the system that they optimized is a complementary assay to the cell culture, that will give a valuable information about the effect of the new tested compound directly on the human enzyme with homology with the targeted enzyme.

Few corrections have been pointed along the revision.

Author Response

Thank you for your very kind comment.
I've fixed everything according to your support. I have carefully revised the concluding paragraph in particular, so I would appreciate it if you could check it out.

Your comment was very valuable. I would be very grateful if you could let me know your email address so I can receive further suggestions.

Best wishes,

Chibana

This manuscript is a resubmission of an earlier submission. The following is a list of the peer review reports and author responses from that submission.

Round 1

Reviewer 1 Report

Nakano et al manuscript reports the development of a reliable evaluation system to evaluate the toxicity of promising drugs targeting Erg25p. The authors also tested two drugs. 

All the manuscript is clearly and carefully written, materials and methods contain the required detail to reproduce the work, results are clearly presented and described. The experimental work appears to have been carried out well. I suggest that it is accepted for publication after the following revisions:

Line 14: “which is greater essentiality than the targets of existing drugs, ”, please rephrase

Line 29: “These resistant bacteria include drug-resistant Candida [1–3]. ” please rephrase, Candida spp. are not bacteria, but yeasts

Line 34-36: “Of the four classes of antifungal agents for invasive candidiasis, polyenes act directly on ergosterol[7,8], while azoles inhibit lanosterol 14α-demethylase (Erg11p)[9–12]. Two of the classes target ergosterol and its biosynthetic pathway. Although not indi-”. There is a lack of link between the sentences. The speech is not fluid.

Line 40: “molecules, and additional candidate target molecules might remain ” please rephrase

Line 50 to 67: the introduction is large. This paragraph might be removed. Line 65-67: “During the cells are depleted of ergosterol, they can grow since the replacement of endogenous 66 ergosterol by uptake of external cholesterol [35,36]” please rephrase.

Line 73: SD medium. Does SD stands for Synthetic Defined? Does YPD stands for Yeast Extract–Peptone–Dextrose? It should be mentioned in the citation.

Line 183 to 196: Generally, the liquid growth assays are performed in Erlenmeyers under a good agitation to ensure a good oxygenation. However, the curves obtained by the authors are OK.

Line 201: “Adjust-ed”

Line 205: “MIC” First citation: minimal inhibitory concentration (MIC)

Line 206: “Determination of MIC against 1181-0519 ” -> “Determination of the MIC of 1181-0519 ” The MIC is a concentration

Line 212: Do the author included a control with DMSO to test its toxicity?

Line 213: “bacterial solution” Candida sp. and Saccharomyces sp. are yeasts. 

Line 215: “The drug was added by mixing 20 ul in the first row and further diluted by 100ul in 2-fold series (the highest drug concentration was 1/10 of the added concentration). 48 h incubation was started at 28 °C. ” Prior to de addition of the 20uL, the first row contained 180uL of water (diluition 1:10)? Then the authors removed 100uL to the next row that contained 100ul of water (diluition 1:2)? The problem is that in line 213, the authors referred that they added the yeast suspension. What did they add first? The yeast suspension or they performed first the drug diluition?

Line 220: “growth of the growth target” or “growth of the control”?

Line 251: IC50 or IG50? The assay is a growth inhibition.

Fig 2, Table 2 and 3: In Fig 2 and Table 2, the results are expressed in uM but in Table 3 they are expressed in ug/mL. The caption of table 3 presented a conversion (1 uM= 0,25 ug/mL) but some readers might not be aware or if they are, they have to perform the conversion. All the results must be expressed in the same units. 

Line 296: “C4-methyl sterol monooxygenase (Erg25p), which is essential for the growth of S. cerevisiae[55] and C. glabrata[16], is a promising target molecule for antifungal drugs with higher efficacy than Erg11p, the target molecule of azoles [16]. ” The enzyme is not only important for S. cerevisiae and C. glabrata.

Line 297: “with higher efficacy” or with higher potential?

Line 318 – 319: “Since the three histidine motifs were conserved (Figure. S4), suggesting that the binding domain of 1181-0519 is other than the histidine motifs”: please rephrase

Fig S3 is unformatted and in the caption A) Penicillium is not in italic as it should.

The manuscript can be checked by a native English speaker.

Author Response

Thank you for your valuable comments and suggestions. We conducted several experiments and endeavored to address them to the best of our ability. In line with the reviewers' suggestions, the revised version eliminated redundant statements, particularly in the introductory section and discussion. As a result, some specific sentences that were identified by each reviewer as requiring revision have been removed. Below, your comments and suggestions have been assigned numbers (Q), and the responses to each comment (A) have been individually numbered to correspond to the relevant (Q).

Q1: Line 14: “which is greater essentiality than the targets of existing drugs, ”, please rephrase

A1: Following the reviewer suggestion, we changed the text to “which is a greater essential target than that of existing drugs,”

Q2: Line 29: “These resistant bacteria include drug-resistant Candida [1–3]. ” please rephrase, Candida spp. are not bacteria, but yeasts

A2: Since contains bacteria and fungi aloso, it was fixed Bactria to pathogen.

Q3: Line 34-36: “Of the four classes of antifungal agents for invasive candidiasis, polyenes act directly on ergosterol [7,8], while azoles inhibit lanosterol 14α-demethylase (Erg11p) [9–12]. Two of the classes target ergosterol and its biosynthetic pathway. Although not indi-”. There is a lack of link between the sentences. The speech is not fluid.

A3: Revised according to the reviewer suggestion.

Q4: Line 40: “molecules, and additional candidate target molecules might remain ” please rephrase

A4: This has been corrected as the result of significant revisions made to the Introduction.

Q5: Line 50 to 67: the introduction is large. This paragraph might be removed. Line 65-67: “During the cells are depleted of ergosterol, they can grow since the replacement of endogenous 66 ergosterol by uptake of external cholesterol [35,36]” please rephrase.

A5: The entire introduction has been revised, including the parts you pointed out.

Q6: Line 73: SD medium. Does SD stands for Synthetic Defined? Does YPD stands for Yeast Extract–Peptone–Dextrose? It should be mentioned in the citation.

A6: As the reviewer pointed out, SD is an abbreviation for Synthetic Defined, and YPD is an abbreviation for Yeast extract, Polypepton and Dextrose. Details of these are described in Material and method.

Q7: Line 183 to 196: Generally, the liquid growth assays are performed in Erlenmeyers under a good agitation to ensure a good oxygenation. However, the curves obtained by the authors are OK.

A7: The automatic meter was programmed to stir at 1,000 rpm for 10 seconds every 10 minutes. I added this to material and method.

Q8: Line 201: “Adjust-ed”

A8: Fixed.

Q9: Line 205: “MIC” First citation: minimal inhibitory concentration (MIC)

A9: Revised according to the reviewer suggestion.

Q10: Line 206: “Determination of MIC against 1181-0519 ” -> “Determination of the MIC of 1181-0519 ” The MIC is a concentration

A10: Followed the suggestion of the reviewers.

Q11: Line 212: Do the author included a control with DMSO to test its toxicity?

A11: DMSO was included. No effect of DMSO could be detected.

Q12: Line 213: “bacterial solution” Candida sp. and Saccharomyces sp. are yeasts. 

A12: Fixed.

Q13: Line 215: “The drug was added by mixing 20 µL in the first row and further diluted by 100ul in 2-fold series (the highest drug concentration was 1/10 of the added concentration). 48 h incubation was started at 28 °C. ” Prior to de addition of the 20uL, the first row contained 180uL of water (diluition 1:10)? Then the authors removed 100uL to the next row that contained 100ul of water (diluition 1:2)? The problem is that in line 213, the authors referred that they added the yeast suspension. What did they add first? The yeast suspension or they performed first the drug diluition?

A13: The text in this paragraph was unclear and has been fixed.

Q13: Line 220: “growth of the growth target” or “growth of the control”?

A13: Fixed according to the reviewer suggestion.

Q14: Line 251: IC50 or IG50? The assay is a growth inhibition.

A14: Avoid inviting doubts from reviewers, half maximal (50%) inhibitory concentration (IC50) was defined in Materials and Methods.

Q15: Fig 2, Table 2 and 3: In Fig 2 and Table 2, the results are expressed in uM but in Table 3 they are expressed in ug/mL. The caption of table 3 presented a conversion (1 uM= 0,25 ug/mL) but some readers might not be aware or if they are, they have to perform the conversion. All the results must be expressed in the same units. 

A15: Following reviewer suggestion, it was unified to µM.

Q16: Line 296: “C4-methyl sterol monooxygenase (Erg25p), which is essential for the growth of S. cerevisiae [55] and C. glabrata [16], is a promising target molecule for antifungal drugs with higher efficacy than Erg11p, the target molecule of azoles [16]. ” The enzyme is not only important for S. cerevisiae and C. glabrata.

A16: Added literature supporting the essentiality of ERG25 in C. albicans.

Q17: Line 297: “with higher efficacy” or with higher potential?

A17: “with higher potential.” This has been corrected to avoid misunderstandings.

Q18: Line 318 – 319: “Since the three histidine motifs were conserved (Figure. S4), suggesting that the binding domain of 1181-0519 is other than the histidine motifs”: please rephrase.

A18: Revised according to reviewer suggestion.

Q19: Fig S3 is unformatted and in the caption A) Penicillium is not in italic as it should.

A19: Revised according to reviewer suggestion.

Reviewer 2 Report

The authors confirmed the functional complementation of C. glabrata ERG25 and human SC4MOL genes after knock-in of ERG25-deficient S. cerevisiae. Then, they used the knock-in strains to evaluate the selective toxicity of a known Erg25p inhibitor. Their findings are important for screening of anti-pathogen compounds and selective toxicity assays. I provided some comments for the authors to consider as outlined below.

General comments:

1) More information is provided in the Introduction. It should be compressed, especially, in the Line 68-112.

2) These are no information of Statistical analysis in the Material and Methods.

3) More results of complementarity of hERG25 and CgERG25 in S. cerevisiae, related to these molecular manipulations, should perform.

4) All the test strains are susceptible to 1181-0519. How about the IC50s and MIC of test strains to polyenes and azoles? And How about the levels of 1181-0519 are clinically achievable without toxicity?

5) These is repeat results describe in the DISCUSSION. Therefore, the part should be revised.

Other comments:

1. Line 24-25: The “AMR” and “MSMO1” should provide full name. “C. albicans” should be Candida albicans. “Candida parapsilosis” should be C. parapsilosis.

2. Line 29: “These resistant bacteria…” bacteria is not including fungi. Therefore, the world of “bacteria” should be revised.

3. Line 140: “Escherichia coli… .” should be italic. All the species, the world of “in vivo” and “in vitro” should be italic. Please revise the issue throughout the manuscript.

4. Line 251: The definition of IC50 should be provided.

Author Response

Thank you for your valuable comments and suggestions. We conducted several experiments and endeavored to address them to the best of our ability. In line with the reviewers' suggestions, the revised version eliminated redundant statements, particularly in the introductory section and discussion. As a result, some specific sentences that were identified by each reviewer as requiring revision have been removed. Below, your comments and suggestions have been assigned numbers (Q), and the responses to each comment (A) have been individually numbered to correspond to the relevant (Q).

General comments:

Q1) More information is provided in the Introduction. It should be compressed, especially, in the Line 68-112.

A1) According to reviewer suggestion, the introduction has been compressed.

Q2) These are no information of Statistical analysis in the Material and Methods.

A2) Statistical analysis was added to Materials and Methods.

Q3) More results of complementarity of hERG25 and CgERG25 in S. cerevisiae, related to these molecular manipulations, should perform.

A3) The presence or absence of endogenous S. cerevisiae ERG25 was demonstrated by PCR in Figure S3.

Q4) All the test strains are susceptible to 1181-0519. How about the IC50s and MIC of test strains to polyenes and azoles? And How about the levels of 1181-0519 are clinically achievable without toxicity?

A4) Susceptibility tests to polyenes and azoles were performed and added (Figure S4). The results revealed no significant differences among the three strains: BY4741, Sc(CgERG25), and Sc(hERG25). However, a significant difference was observed for 1181-0519. Toxicity tests on cultured cells were conducted and added to Figure 3. The results demonstrated no cytotoxicity, even at concentrations as high as 100 µM of 1181-0519. Regarding clinical applications, it is challenging to provide a definitive assessment without conducting pharmacokinetic studies using animals. Nevertheless, it is presumed that a concentration of 2 µM (equivalent to 0.5 µg/mL) is clinically achievable.

Q5) These is repeat results describe in the DISCUSSION. Therefore, the part should be revised.

A5) Duplicate parts in the results and the discussion have been shortened.

Other comments:

Q1. Line 24-25: The “AMR” and “MSMO1” should provide full name. “C. albicans” should be Candida albicans. “Candida parapsilosis” should be C. parapsilosis.

A1. Following reviewer suggestion, AMR and MSMO1 have been given their full names, and the description of Candida species has been revised.

Q2. Line 29: “These resistant bacteria…” bacteria is not including fungi. Therefore, the world of “bacteria” should be revised.

A2. Since contains bacteria and fungi also, it was fixed Bactria to pathogen.

Q3. Line 140: “Escherichia coli… .” should be italic. All the species, the world of “in vivo” and “in vitro” should be italic. Please revise the issue throughout the manuscript.

A3. Following the reviewer’s suggestion, “in vivo” and “in vitro” have been italicized.

Q4. Line 251: The definition of IC50 should be provided.

A4. IC50 was defined in lines 143-145 according to the reviewer's suggestion.

Reviewer 3 Report

Nakano et al. J of Fungi, ERG25

The authors seek to exploit the ERG25 protein in the ergosterol synthesis pathway as an important antifungal target. They have chosen this protein based upon their statement that ERG25 protein has a greater essentiality as an antifungal target.

Comments on text organization

Overall, there are several textual mistakes that need to be changed, mostly regarding sentence construction. I have listed page numbers where changes are needed.

Lines 13-14

Lines 65-66

Lines 70-71

Lines 225-226

Line 322, Meaning of “enhancing broadening.”

Line 349, remove “In part of”

Line 375, meaning of “drop-out risks.”

Also. there are two types of text alignments throughout the manuscript: (1) alignments (sentences) start on the left and are not aligned on the right for some sections (2) alignments are “justified”, aligned equally on left and right borders. Convert the entire text to one or the other.  It sort of resembles a “cut and paste” effect. Was this used to provide text that was originally cited in other publications by this group?

Also, font size varies in text sections. Please change the entire text to a single font size and spacing concurreance.

An example of this is page 3, the upper half being justified, the lower half being left alignment and spacing differences. I hope this does not reflect a “Cut and paste effort.”

Bottom of page 4. The text ends at the very bottom of this page. Remove and add it to the top of page 5.

Introduction

Again, the font size, spacing etc is different in sections of the Introduction.  Is this due to a copy and paste usage?

The Introduction is much too long, and more focus needs to be on the ERG25. I would think references to other Ergs can be summarized and references indicated.

Also, the spacing is of two types.

In summary of the introduction, I feel the Introduction is not cleanly presented and requires major changes.

Data

Figure 1 legend, line 239. The averages of three replicates were plotted “per strain”.  Also, most of the legend has been repeated from the methods section.

Figure 1 data is reasonable.

Figure 2B.  Figure markers for each strain should be different.

Table 2. Is fine, but my question relates to ERG25 transcription levels of each strain.  This data needs to be included.

Page 3 again, there are two fonts, alignments, and spacings.

Table 3, C. tropicalis:  is the 4 ug/ml to the compound an intermediate inhibitory concentration? If so, an intermediate MIC can refer to resistance.

Conclusion. The authors make a point that the toxicity of 1181-0519 can be more readily evaluated using their “knocked in” methodology. I would agree, but the compound still must be tested in other ways.  One of the simplest is an in vitro assay of the compound with human cell Iines. I am not asking that the authors to do this, but it should be mentioned as a next step in the process. Other assays would need to be followed.

I have listed  several sections that can be improved to restore text consistency. Spacing of sentences, font size, and alignment of text need to be redone.

Author Response

Thank you for your valuable comments and suggestions. We conducted several experiments and endeavored to address them to the best of our ability. In line with the reviewers' suggestions, the revised version eliminated redundant statements, particularly in the introductory section and discussion. As a result, some specific sentences that were identified by each reviewer as requiring revision have been removed. Below, your comments and suggestions have been assigned numbers (Q), and the responses to each comment (A) have been individually numbered to correspond to the relevant (Q).

Q1) Lines 13-14

A1) Revised according to the reviewer’s suggestion.

Q2) Lines 65-66

A2) This has been corrected due to the compression of the introduction.

Q3) Lines 70-71

A3) This has been corrected due to the compression of the introduction as well.

Q4) Lines 225-226

A4) Revised according to the reviewer’s suggestion.

Q5) Line 322, Meaning of “enhancing broadening.”

A5) Revised according to the reviewer’s suggestion.

Q6) Line 349, remove “In part of”

A6) Revised according to the reviewer’s suggestion.

Q7) Line 375, meaning of “drop-out risks.”

A7) Line 345-347 on the revised manuscript, drop-out risk was used to mean “removing a candidate compound from the development process at this stage can result in significant losses, as previous investments would be wasted. ” but the phrase of drop-out risk has been removed to avoid ambiguity.

Q8) Also. there are two types of text alignments throughout the manuscript: (1) alignments (sentences) start on the left and are not aligned on the right for some sections (2) alignments are “justified”, aligned equally on left and right borders. Convert the entire text to one or the other.  It sort of resembles a “cut and paste” effect. Was this used to provide text that was originally cited in other publications by this group?

Also, font size varies in text sections. Please change the entire text to a single font size and spacing concurreance.

An example of this is page 3, the upper half being justified, the lower half being left alignment and spacing differences. I hope this does not reflect a “Cut and paste effort.”

A8) It seems like to happen the settings to match what you submitted for English proofreading and the parts you made changes to afterwards. Additionally, it seems that the two-sided portion on the three pages that was pointed out by the reviewer occurred when the manuscript was composed using different software and combined into a single submitted manuscript. In other words, it appears that the difference in font size in the text section occurred because we ourselves cut and pasted the text we created in a different format. We aligned these differences with the spacing according to reviewer suggestion.

Q9) Bottom of page 4. The text ends at the very bottom of this page. Remove and add it to the top of page 5.

A9) Revised according to the reviewer’s suggestion.

Introduction

Q10) Again, the font size, spacing etc is different in sections of the Introduction.  Is this due to a copy and paste usage?

The Introduction is much too long, and more focus needs to be on the ERG25. I would think references to other Ergs can be summarized and references indicated.

Also, the spacing is of two types.

A10) According to the reviewer’s suggestions. Line spacing has been standardized.

Q11) In summary of the introduction, I feel the Introduction is not cleanly presented and requires major changes.

A11) The introduction has been significantly revised following reviewer suggestion.

Data

Q12) Figure 1 legend, line 239. The averages of three replicates were plotted “per strain”.  Also, most of the legend has been repeated from the methods section.

A12) The legend of figure 1 has been revised following reviewer suggestion.

Figure 1 data is reasonable.

Q13) Figure 2B.  Figure markers for each strain should be different.

A13) Markers were revised as per the reviewer's suggestion.

Q14) Table 2. Is fine, but my question relates to ERG25 transcription levels of each strain.  This data needs to be included.

Q14) We attempted RNA sequencing. However, it was not successful due to cost constraints.

Q15) Page 3 again, there are two fonts, alignments, and spacings.

A15) Revised according to the reviewer’s suggestion.

Q16) Table 3, C. tropicalis:  is the 4 ug/ml to the compound an intermediate inhibitory concentration? If so, an intermediate MIC can refer to resistance.

A16) As pointed out by the reviewer, among these Candidia species, C. tropicalis showed low susceptibility to 1181-0519. However, we have not made a definitive determination as to whether this is resistant or not, as it is impossible to determine whether this is resistant without investigating more species and strains.

Q17) Conclusion. The authors make a point that the toxicity of 1181-0519 can be more readily evaluated using their “knocked in” methodology. I would agree, but the compound still must be tested in other ways. One of the simplest is an in vitro assay of the compound with human cell Iines. I am not asking that the authors to do this, but it should be mentioned as a next step in the process. Other assays would need to be followed.

A17) As recommended by the reviewers, toxicity tests on cultured cells were performed and the results have been added to Figure 3. The results showed no cytotoxicity even at concentrations as high as 100 µM 1181-0519.

Comments on the Quality of English Language

Q18) I have listed several sections that can be improved to restore text consistency. Spacing of sentences, font size, and alignment of text need to be redone.

A18) Revised according to the reviewer’s suggestion.